# A Diet Induced Maladaptive Increase in Hepatic Mitochondrial DNA Precedes OXPHOS Defects and May Contribute to Non-Alcoholic Fatty Liver Disease

**DOI:** 10.3390/cells8101222

**Published:** 2019-10-08

**Authors:** Afshan N Malik, Inês C.M. Simões, Hannah S. Rosa, Safa Khan, Agnieszka Karkucinska-Wieckowska, Mariusz R. Wieckowski

**Affiliations:** 1Department of Diabetes, School of Life Course, Faculty of Life Sciences and Medicine, King’s College, London SE1 1UL, UK; hannah.rosa@kcl.ac.uk (H.S.R.); safa.k.khan@kcl.ac.uk (S.K.); 2Nencki Institute of Experimental Biology, Polish Academy of Sciences, Pasteur 3 Str., 02-093 Warsaw, Poland; i.simoes@nencki.gov.pl; 3Department of Pathology, The Children’s Memorial Health Institute, 04-730 Warsaw, Poland; A.Karkucinska-Wieckowska@IPCZD.pl

**Keywords:** non-alcoholic fatty liver disease (NAFLD), steatosis, mitochondria, mitochondrial DNA (mtDNA), oxidative phosphorylation (OXPHOS)

## Abstract

Non-alcoholic fatty liver disease (NAFLD), an increasingly prevalent and underdiagnosed disease, is postulated to be caused by hepatic fat mediated pathological mechanisms. Mitochondrial dysfunction is proposed to be involved, but it is not known whether this is a pathological driver or a consequence of NAFLD. We postulate that changes to liver mitochondrial DNA (mtDNA) are an early event that precedes mitochondrial dysfunction and irreversible liver damage. To test this hypothesis, we evaluated the impact of diet on liver steatosis, hepatic mtDNA content, and levels of key mitochondrial proteins. Liver tissues from C57BL/6 mice fed with high fat (HF) diet (HFD) and Western diet (WD, high fat and high sugar) for 16 weeks were used. Steatosis/fibrosis were assessed using haematoxylin and eosin (H&E) Oil Red and Masson’s trichome staining and collagen content. Total DNA was isolated, and mtDNA content was determined by quantifying absolute mtDNA copy number/cell using quantitative PCR. Selected mitochondrial proteins were analysed from a proteomics screen. As expected, both HFD and WD resulted in steatosis. Mouse liver contained a high mtDNA content (3617 ± 233 copies per cell), which significantly increased in HFD diet, but this increase was not functional, as indicated by changes in mitochondrial proteins. In the WD fed mice, liver dysfunction was accelerated alongside downregulation of mitochondrial oxidative phosphorylation (OXPHOS) and mtDNA replication machinery as well as upregulation of mtDNA-induced inflammatory pathways. These results demonstrate that diet induced changes in liver mtDNA can occur in a relatively short time; whether these contribute directly or indirectly to subsequent mitochondrial dysfunction and the development of NAFLD remains to be determined. If this hypothesis can be substantiated, then strategies to prevent mtDNA damage in the liver may be needed to prevent development and progression of NAFLD.

## 1. Introduction

Non-alcoholic fatty liver disease (NAFLD) is an increasingly prevalent and underdiagnosed disease, which is estimated to affect a staggering 25% of the population [1]. NAFLD is strongly associated with diabetes, obesity, and metabolic syndrome and is believed to be caused by diet related mechanisms that lead to the accumulation of hepatic fat [2]. The disease progresses over a long period, sometimes decades, through defined clinical stages. In the early stages, often in the presence of insulin resistance, liver steatosis, the accumulation of fat droplets in the liver, develops due to decreased fatty acid oxidation and increased fat uptake. This initial clinically silent but potentially reversible steatosis progresses to a less reversible and clinically silent stage known as non-alcoholic steatohepatitis (NASH) during which a number of changes take place, for example; oxidative stress, lipid peroxidation, and proinflammatory cytokine activation, which further compromise liver function [3]. NASH eventually progresses to irreversible cirrhosis, permanent tissue scarring and damage [4,5] and, in some cases, liver carcinoma. Even though the rate of developing liver cancer or requiring transplant is relatively low compared to other forms of liver disease, the sheer scale of the number of people with NASH makes it a huge problem [6]. With the increasing worldwide prevalence of NAFLD, there is a strong clinical and economic need to identify biomarkers before irreversible organ damage occurs and to understand the early mechanisms in order to design preventative therapeutic strategies.

Mitochondria are eukaryotic cellular organelles that are the major site of energy production by oxidative phosphorylation (OXPHOS), producing ATP to support cellular energy-requiring functions [7]. The liver requires healthy functional mitochondria to undertake the many functions attributed to this large organ, including metabolism of fats, proteins, and carbohydrates, storage of glycogen, and blood detoxification to name only a fraction [8]. The liver contains millions of hepatocytes, cells with a life span of ~150 days, and over half of the body’s supply of macrophages, known as Kupffer cells, needed to fight infections in the body [9,10]. Maintenance of a healthy level of these cells is another energy requiring function of the liver. Increasing evidence suggests that impaired mitochondrial function plays a major role in the development of diet-induced NAFLD, as recently reviewed by Simões et al. [11]. However, the exact mechanisms by which diet-related changes lead to hepatic mitochondrial dysfunction and their temporal relationship to different stages of NAFLD have not been delineated.

Mitochondria contain their own extranuclear circular genome known as mitochondrial DNA (mtDNA), a small circular molecule of 16.5 kb present in multiple copies in cells [12,13]. MtDNA content varies between different cell types depending on the cell’s bioenergetic needs but can also change in response to physiological stimuli, leading to the proposal that mtDNA could be a biomarker of mitochondrial dysfunction [14,15]. Whilst the quantity of mtDNA can easily be measured from small amounts of biological samples using quantitative polymerase chain reaction (qPCR), methodological issues such as the co-amplification of nuclear mitochondrial insertion sequences (NUMTs) and dilution bias [16] can result in errors. To our knowledge, there are very few reports of absolute mtDNA quantity measured as mtDNA copy numbers per cell from liver tissue from human or animal models to date, and therefore the range of mtDNA content in normal liver remains to be established [17]. Previously, some studies have reported reduced mtDNA in NAFLD and NASH whilst measuring the mtDNA content as a relative arbitrary number rather than exact quantities [18,19].

Given that mtDNA levels correspond to the bioenergetic requirements of cells and tissues, and that temporal changes in mtDNA during disease can provide important insight into the underlying mechanisms, it is important to determine normal hepatic mtDNA levels to establish whether these levels fluctuate during the progression of NAFLD and whether these changes have an impact on mitochondrial function. In the current paper, our aim was to determine the normal range of mtDNA copy number in mouse liver to examine whether high fat (HFD) or Western diet (WD)—the latter containing both high fat and high sugar—can affect this normal range during the development of steatosis.

## 2. Materials and Methods

The studies presented in the paper were approved by Local Ethical Committees (Resolution No. 200/2016 on 11 December 2016) and performed in accordance with the guidelines based on national laws that are in full agreement with the European Union directive on animal experimentation.

### 2.1. Animal Model and Diet

C57BL/6 male mice were used for this study conducted over a period of 16 weeks during which mice were fed ad libitum and housed at temperatures between 21–23 °C in 50–60% humidity (10–15 exchanges of air per hour). Mice were fed one of the following 3 diets:

a) Control (C)—animals fed with standard chow diet (ssniff R/M);

b) High-fat (HFD)—animals fed with a high-fat diet [ssniff EF R/M with 30% fat composed of long chain saturated fatty acids (7.49% C16:0 (palmitic acid), 5.29% C18:0 (stearic acid), and 10.98% C18:1 (oleic acid))];

d) Western diet (WD)—animals fed with high-fat diet (ssniff EF R/M with 30% fat composed of long chain saturated fatty acids) and high-sucrose content (30%) in the drinking water.

Further details of the composition of these diets are available in Appendix A.

The diet treatment started at 8 weeks old (2 months), and the treatment finished at 24 weeks (6 months), at which point they were subjected to isoflurane inhalation anesthesia and then sacrificed by cervical dislocation. For haematoxylin and eosin (H&E) and Masson Trichrome staining, livers were fixed in 10% phosphate buffered formalin for 24–48 h at room temperature. Fixed livers were trimmed into appropriate size and shape and placed in embedding cassettes. The paraffin embedding schedule was as follows (total 16 h): 70% ethanol, two changes, 1 h each 80% ethanol, one change, 1 h 95% ethanol, one change, 1 h 100% ethanol, three changes, 1.5 h each xylene, three changes, 1.5 h each paraffin wax (58–60 °C), two changes, 2 h each embedding tissues into paraffin blocks.

For Oil Red staining, livers were placed in tissue embedding medium (matrix) and frozen in isopentane cooled by liquid nitrogen.

### 2.2. Liver Histology

a) H&E and Masson Trichrome staining: Paraffin blocks were cut into 3 µm thick slices and mounted on SuperFrost microscope slides (from Gerhard Menzel GMBH, Braunschweig, D-38116 Germany). After step-wise deparaffinization and rehydration, slides were stained with haematoxylin and eosin according to standard protocols. The standard protocol of the manufacturer for Masson Trichrome staining with the use of Weigert’s iron haematoxylin solution Set and Trichrome Stain (Masson) Kit (both from Sigma-Aldrich Sp. z o.o., 61-626 Poznan, Poland) was followed [20].

b) Oil Red staining: Frozen in tissue embedding medium, livers were cut into 8 µm thick slices and mounted on SuperFrost Plus microscope slides (from Gerhard Menzel GMBH, Braunschweig, D-38116 Germany). Neutral lipid accumulation in livers from chow, HFD, and WD fed mice were visualized with the use of Oil Red staining [20].

### 2.3. Mitochondrial DNA Quantification

Total genomic DNA was extracted from frozen liver sections (1mm × 1mm × 1mm) using the DNeasy blood and tissue kit according to the manufacturer’s guidelines (Qiagen, Manchester, UK). DNA was diluted to 10 ng/uL and treated by sonication to minimise effects of dilution bias [16]. Absolute mtDNA copy number was determined by qPCR utilising primers for specific mitochondrial and nuclear genome targets (mMitoF1/R1 and mB2MF1/R1 respectively), as detailed in Malik et al. [17]. Sample DNA was loaded in triplicate alongside a 5-point standard curve consisting of primer-specific amplicons of known copy number. The ratio of mtDNA to *B2m* copies (MtN) was multiplied by 2 to calculate the mtDNA content per cell.

### 2.4. Proteomics

For proteomics analysis, control mice fed with standard chow diet, mice fed with HFD, and mice fed with WD (n = 3 per condition) were sacrificed by cervical dislocation. Livers were isolated, extensively washed in phosphate buffered saline (PBS), and stored in liquid nitrogen. Liquid chromatography-MS3 spectrometry (LC-MS/MS) was carried out at the Thermo Fisher Center for Multiplexed Proteomics (Dept. of Cell Biology, Harvard Medical School, Cambridge, MA, USA). Peptide fractions were analyzed using an LC-MS3 data collection strategy on an Orbitrap Fusion mass spectrometer (Thermo Fisher Scientific Inc., Waltham, MA, USA). Further details of methodology can be found in Appendix A.

### 2.5. Statistical Analysis

Statistical analyses of collagen protein levels, mtDNA copy number, and proteomics datasets were performed using GraphPad Prism version 8.0.0 for Windows (GraphPad Software, San Diego, CA, USA). Normality was determined by the Shapiro–Wilk normality test, and non-normally distributed datasets were log-transformed before analysis. Parametric tests such as student t-test and one-way ANOVA were performed on normally distributed and log-transformed datasets. Significant differences are denoted where *p* > 0.05.

## 3. Results

### 3.1. Development of Diet Induced Liver Steatosis

C57BL/6 mice were fed one of three diets (control, HFD, and WD) for a period of 16 weeks. Histological examination of the liver sections after this period showed increased accumulation of fat (in the form of lipid droplets) in the hepatocytes of mice fed with both treatments relative to the controls (Figure 1, H&E and Oil Red panels). Fat accumulation (steatosis) could be observed, suggesting that there was diet induced liver dysfunction. Both HFD and WD led to the development of steatosis with accelerated development of micro and macro steatosis and a predominance of larger fat droplets evident in the WD liver (Figure 1, Oil Red panel). Moreover, in the WD livers, perivascular fibrosis was detected (Figure 1, Masson Trichrome panel), suggesting more advanced liver damage than in the HFD.

The occurrence of more advanced liver damage in WD compared to HFD fed mice was further corroborated by the increased evidence of fibrosis. The levels of collagens V and VI (early markers of fibrosis) were significantly increased in WD versus both HFD and controls, supporting the view that WD liver showed a more advanced stage of disease (Figure 2).

### 3.2. Mitochondrial DNA Content in the Control Mouse Liver

Determination of absolute copy numbers of liver mtDNA from the control mice showed that the liver had a relatively high mtDNA content, and there was little variation in total mtDNA content in the different control mice (Table 1). The values of mtDNA content ranged from 3472 to 3885 copies per cell, with mean values for three control animals of 3617 ± 233 copies of mtDNA per nuclear genome. These data show that, on average, the liver has an excess of 3500 copies of the mitochondrial genome per cell.

### 3.3. Hepatic Mitochondrial DNA Content Increased in Parallel with Steatosis

In mice fed with the HFD, liver mtDNA content per cell ranged between 4642 and 6495 (mean ± SD = 5395 ± 975). This represented a significant increase of >50% in mtDNA content relative to the controls (Table 1, Figure 3). However, in the WD fed animals, we saw a decrease relative to the amount seen in HFD mice, which was not quite significant (*p* = 0.076); the levels seemed to resemble those seen in the control group (mean ± SD = 3822 ± 600, n = 3).

### 3.4. Alterations in Hepatic Mitochondrial Proteins Suggest Reduced OXPHOS, Reduced Mitochondrial Biogenesis, and Increased Inflammation

Our data above showed increased mtDNA in HFD but not in WD (Figure 3), which suggested that the initial increase in HFD may be a maladaptive response that is subsequently reduced in more advanced disease. To explore this further, we used proteomics to measure certain proteins within the same liver sections. We wanted to determine if the mtDNA increase in HFD liver was functional, in which case we would expect an increase in mtDNA encoded OXPHOS proteins, or whether the increase was not functional, which could then cause dysregulation of mitochondrial biogenesis and mtDNA-mediated inflammation. Levels of proteins of interest from chow, high-fat, and Western diet fed mice are shown in Appendix A.

#### 3.4.1. Reduced mtDNA Encoded OXPHOS Proteins in Steatotic Liver

Since we observed increased mtDNA in the liver of HFD mice, we expected to see an increase in mtDNA encoded OXPHOS subunits. In HFD mice livers, measurement of seven mtDNA encoded subunits showed that only one (MTND1: NADH dehydrogenase 1, component of complex 1) showed a significant increase, whereas none of the others were changed relative to controls. In WD livers, none of the OXPHOS subunits encoded by mtDNA were increased, and instead we saw a significant decrease in two of them (MTND2; NADH dehydrogenase 2, complex 1; and MTCO3: Cytochrome C Oxidase III, complex IV) (Figure 4a).

#### 3.4.2. Reduced Nuclear Encoded OXPHOS Proteins in Steatotic Liver

We examined five proteins—one for each of the five OXPHOS complexes—to see if their levels were altered during steatosis. In HFD fed mice, there was an increase in two proteins (NDUFB8 (NADH:ubiquinone oxidoreductase subunit B8) of complex 1 and COX5B (Cytochrome C Oxidase Subunit 5B) of complex IV); note that the increase in complex 1 subunit correlated with the mtDNA encoded complex 1 subunit MTND1, which was also increased. Therefore, it is possible that, at first, mtDNA increase was accompanied by increased transcription and increased translation of some subunits. However, we also saw a modest but significant decrease in SDHA (Succinate Dehydrogenase Complex Flavoprotein Subunit A, complex 2) in HF liver, which could indicate a dysregulation of OXPHOS machinery synthesis. In WD mice livers, there was a consistent decrease in four out of five of the subunits we examined, supporting the trend of reduction seen above (Figure 4b).

#### 3.4.3. Reduction in Proteins Involved in mtDNA Replication in Steatosis

As we saw an increase in hepatic mtDNA in HFD mice, we examined some of the proteins involved in mtDNA replication machinery to determine if this increase was caused by increased mtDNA replication (Figure 5a). However, instead of an increase, we saw a reduction in two of these proteins in the HFD diet fed mice alongside a modest increase in mitochondrial transcription factor A (TFAM) levels. In WD mice livers, we saw a significant reduction in three proteins of the mtDNA replication machinery (Figure 5b).

#### 3.4.4. Alterations in Steatotic Livers Proteins Suggests Altered Nuclear Genome Methylation and Increased Inflammation

There was no obvious change in proteins involved in mitochondrial dynamics in HFD alone, but in WD, we saw a reduction in mitofusin 1 (MFN1), which is part of the fusion machinery of the mitochondrial network (Figure 5c). In WD, we saw a significant increase in DNMT1 [DNA (cytosine-5)-methyltransferase 1] (Figure 5d). We also observed a significant increase in NFKB2 (nuclear factor kappa-light-chain-enhancer of activated B cells) in HF alone and an enhanced increase in NFKB2 and NFKB1 in WD, suggesting that, in WD livers, there may be mtDNA-mediated inflammation via the toll-like receptor 9 (TLR9) pathway (Figure 5e).

## 4. Discussion

In the current study, we proposed to determine whether mtDNA changes occur during steatosis in the liver, as the demonstration of this could provide novel information about the mechanisms underlying mitochondrial dysfunction in NAFLD. We previously reported diabetes-induced changes in cellular mtDNA, which seemed to precede and contribute to the disease mechanisms leading to organ dysfunction [21]. We showed then that hyperglycemia could directly cause a >two-fold increase in cellular mtDNA in cultured cells. The increased mtDNA was a maladaptive response—it did not improve mitochondrial function, as it seems to be non-functional, since transcription of the mtDNA was reduced, the mtDNA appeared damaged, and we observed increased inflammation [21,22]. These data led us to propose that there is a maladaptive initial increase of cellular mtDNA during conditions of oxidative stress, which can lead to bioenergetic deficit and increased inflammation [23]. Therefore, in the context of the current study, we speculated that, if we could demonstrate diet-induced changes to liver mtDNA content, this could then provide a strong rationale for further study of the temporal changes in mtDNA during NAFLD pathogenesis and its role in disease progression.

Since mtDNA content can be used as an indicator of mitochondrial content and cellular energy metabolism [24], determination of normal hepatic mtDNA content is an important indicator of the bioenergetic needs to the liver. Despite the liver being utilised as a rich source for isolated mitochondria in bioenergetic and cellular respiration work for many decades [25] and various studies comparing relative mtDNA content in disease conditions, there is surprisingly little information regarding the normal range of mtDNA copies per cell in the liver. Therefore, in the current study, as well as assessing liver mtDNA changes during steatosis, we also sought to determine normal hepatic mtDNA content in mouse liver.

Using absolute quantification to determine the levels, we found that mouse livers contain high levels of cellular mtDNA—our data suggests >3500 mtDNA copies per cell—highlighting the key role of mitochondrial metabolism in liver function. In studies comparing the metabolic rates of different organs in humans, the liver was shown to have a metabolic activity of 200 kcal/kg/day, just below that of the brain at 240 kcal/kg/day [26,27]. Although this is only around half of those reported for heart and kidneys, it is substantially greater than in muscle, emphasising the relatively high metabolic demand of the liver. A similar trend was observed in proteomic analysis of mouse organs showing highest levels of cytochrome c in the heart, followed by the kidneys, the brain, the gut, and the liver [28].

The high amount of mtDNA in liver cells at >3500 copies per cell is an important point when considering the potential impact of altered mtDNA levels on cell, tissue, and organ health. The loss of this amount of mtDNA is likely to affect energy output, and indeed it has been reported that OXPHOS is reduced in NAFLD. Koliaki et al. [29] demonstrated that, in obese individuals, there was a compensatory increase in hepatic mitochondrial respiration but that, over time, the cells lost this capacity, which they described as metabolic flexibility, stating that the loss of metabolic flexibility was associated with increased hepatic insulin resistance and leaking mitochondria. If one considers the amount of mtDNA present per liver cell, leakage of its contents into the cytosol is very likely to lead to inflammation, since mtDNA that has leaked outside of the mitochondria can initiate an inflammatory response by activating the TLR9 receptor upstream to TNF (Tumor necrosis factor) alpha [30,31]. Interestingly, inflammation is a key part of NAFLD pathogenesis and progression.

In the current study, we showed that a high fat diet led to a significant increase in mtDNA in the liver. Although a few studies previously reported liver mtDNA changes in NAFLD, the methods used and the reporting of relative rather than absolute mtDNA copy numbers make it very difficult to compare across studies. As a consequence of such issues, there is little consensus on the relationship between mtDNA content and NAFLD/NASH with conflicting reports of both increase and decrease in mtDNA levels [18,19,32,33] or no change with disease [29]. As circulating cellular and cell-free mtDNA has garnered interest as a potential biomarker for a number of diseases, multiple different methods are being used for sample collection, processing, storage, and DNA extraction and analysis, which leads to highly variable reporting of mtDNA copy numbers [23]. Overall, it is difficult to collate data from multiple sources due to methodological differences and reporting of relative rather than absolute values. Furthermore, many studies of NAFLD and NASH do not account for or report on the status or the impact of diabetes in patients or animal models.

According to our hypothesis, an initial maladaptive increase of mtDNA in steatosis could result in non-functional and damaged mtDNA, which would then be released into the circulation. There are two consequences of this possibility. The first is that there may be increased circulating mtDNA in patients at early stages of NAFLD, which may be utilised as a biomarker at the clinically silent stage of NAFLD, and the second is that elevated mtDNA in circulation could cause inflammation. In support of this idea, we saw dysregulation of mtDNA-encoded OXPHOS proteins in HFD mouse livers and a decrease in OXPHOS proteins encoded by both the mitochondrial and the nuclear genomes in WD mice. The almost 50% reduction in some OXPHOS proteins observed is likely to have had a significant impact on the ability to generate ATP, which was very likely to significantly affect energy output in the liver. These data suggest that, in WD mice, there may have been an earlier maladaptive increase in mtDNA, and by the time we measured at 16 weeks, the mtDNA had become damaged and levels were in decline. Therefore, even though mtDNA levels in WD mice seemed to be the same as the control, they may have been a mixture of normal and damaged/non-functional mtDNA. The current study was focused on understanding the mechanisms underlying early stages of NAFLD (steatosis without progression into a non-steatohepatitis phenotype) and required us to mimic Western diets, which are mostly enriched in fat and sugar. The diet comprised 30% fat (enriched with 7.49% palmitic acid, 5.29% stearic acid, and 10.98% oleic acid) alone (HFD) or in combination with 30% sucrose (WD) present in the drinking water. The choice of sucrose over fructose as the sugar component in the diet was made since fructose is suggested to accelerate liver damage, leading to fast development of fibrosis in NAFLD models [34,35]. Therefore, we considered that the high-fat or the high-fat combined with sucrose as diets, were more likely to yield information on early mechanisms. The exact mechanisms by which high sugar may contribute to NAFLD are not fully understood; however, it has recently become clear that damage to mitochondria is involved, and some of the mechanisms proposed are described in the following review [36]. Interestingly, it is proposed that ER stress and mitochondrial dysfunction underlie the development of fibrosis and inflammation. As such, our study could provide a missing link, since our data suggest that early maladaptive increases in mtDNA may be involved in these processes.

We propose that both early HF and Western diets lead to oxidative stress and a maladaptive increase in hepatic mtDNA, and we predict that this increase is non-functional and that mtDNA transcription and translation could be blocked due to the redox status of the cell, and removal of damaged mtDNA may become impaired. Over time, mtDNA could become slightly damaged, and therefore mutations would begin to accumulate at a low level, eventually reaching the threshold for contributing to either a bioenergetic deficit in some mitochondria or release of contents in others, causing inflammation. Interestingly, in a study by Garcia-Martinez et al. [37], plasma mtDNA levels were significantly increased in obese patients with elevated serum ALT (Alanine aminotransferase) compared to lean and obese controls. This mtDNA was associated with microparticles originating from the liver and capable of causing TLR9-mediated inflammation in cell lines, and therefore it could have originated from the maladaptive increase in liver mtDNA in early stages of disease. The concept of mitochondrial leakage in the liver was previously proposed [38], and if mitochondrial content is being released, then it is likely that the several thousand copies of mtDNA per cell would also be released and could contribute to inflammation. In fact, inflammation in NAFLD is well known, and recently, fatty-acid induced activation of the NLRP3 (NOD-, LRR- and pyrin domain-containing protein 3) inflammasomes in mouse liver Kupffer cells was postulated to be caused by mtDNA release [39]. MtDNA isolated from high fat fed mice elicited a strong TLR9 activtion in mouse Kupffer cells. Handa et al. [40] proposed in a review that mtDNA from hepatocytes might be a ligand for TLR9. In a human study, both plasma and hepatocyte mtDNA levels were increased in NASH patients [29]. Furthermore, it was reported that patients with NAFLD had increased levels of mtDNA damage and heteroplasmy, [38] which we speculate was caused at early stages of steatosis and which contributed to the disease. The data in these studies and the current paper support partial aspects of our hypothesis (Figure 6). However, in order to test this hypothesis fully, studies are needed at defined stages of disease that can detect cellular mtDNA changes in the liver and the cell free mtDNA changes in the periphery over time. If the hypothesis we propose can be substantiated, then strategies to prevent diet induced damage to hepatic mtDNA would be required to prevent the development and the progression of NAFLD.

In the current study, we demonstrated an increase in hepatic mtDNA content during the early stages of steatosis as well as alterations to mRNA species involved in mitochondrial function. While these findings provide partial support for our suggested hypothesis, we are aware of the limitations of the study. Firstly, we recognize the small group sizes used; however, these were sufficient to see statistically significant differences between the chow and the diet groups, particularly with respect to the altered mtDNA content in the HFD mice, suggesting a strong effect of diet on hepatic mtDNA. The exact mechanism for this increase in mtDNA content is currently not understood, and further studies into mitophagy and mitochondrial turnover could elucidate the pathways involved in this potentially maladaptive response. Although we indicated, based on mtDNA copy number changes, that mtDNA damage may be present in the steatotic liver, this was not directly measured. Additionally, the functional associations between mtDNA changes and reactive oxygen species (ROS)/antioxidant imbalance or possible downstream inflammatory and bioenergetic alterations were not investigated here. However, the involvement of aspects of these pathways in the development of NAFLD/NASH has been presented in wider literature.

## 5. Conclusions

In conclusion, our data suggest that early maladaptive changes in hepatic mtDNA content and mitochondrial proteins may precede mitochondrial dysfunction. The potential impact of such changes on the development of fibrosis warrants further investigation.

## Figures and Tables

**Figure 1 cells-08-01222-f001:**
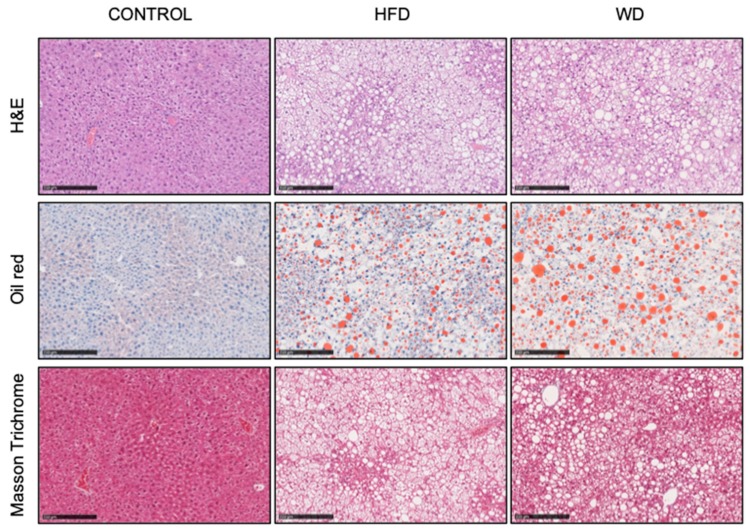
Histological staining of livers from mice fed with standard chow (C), high-fat diet (HFD), and Western (WD) diet at 16 weeks. Haematoxylin and eosin (H&E), Oil Red and Masson Trichrome staining shown in the upper and the lower panels, respectively. Scale bars, 250 μm; magnification is 10×.

**Figure 2 cells-08-01222-f002:**
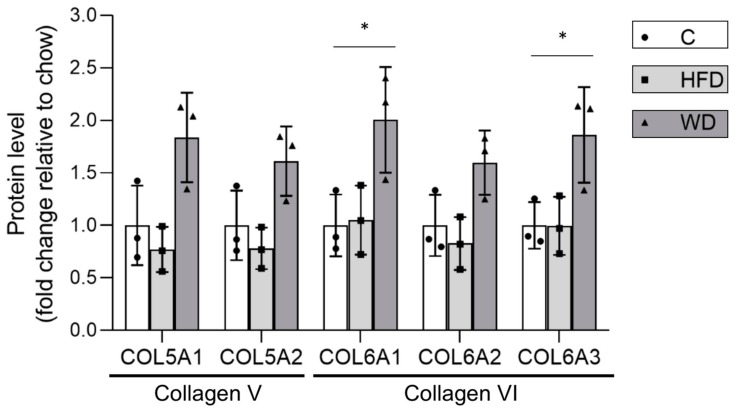
Altered levels of COL5A1, COL5A2 (Collagen Type V Alpha 1 and 2 chains), COL6A1, COL6A2, and COL6A3 (Collagen Type VI Alpha 1, 2 and 3 chains) in steatotic livers of mice fed with standard chow, HFD, and WD. The levels of several collagens are seen to be significantly altered in steatotic livers of mice fed with Western diet, suggesting increased fibrosis. Changes are shown relative to levels in chow-fed mice. Bars indicate mean ± standard deviation for n = 3 mice; individual data points are shown for each mouse fed chow diet (circles, white bars), HFD (squares light grey bars), and WD (triangles, dark grey bars). * *p* < 0.05 as determined by unpaired student t-test versus chow diet.

**Figure 3 cells-08-01222-f003:**
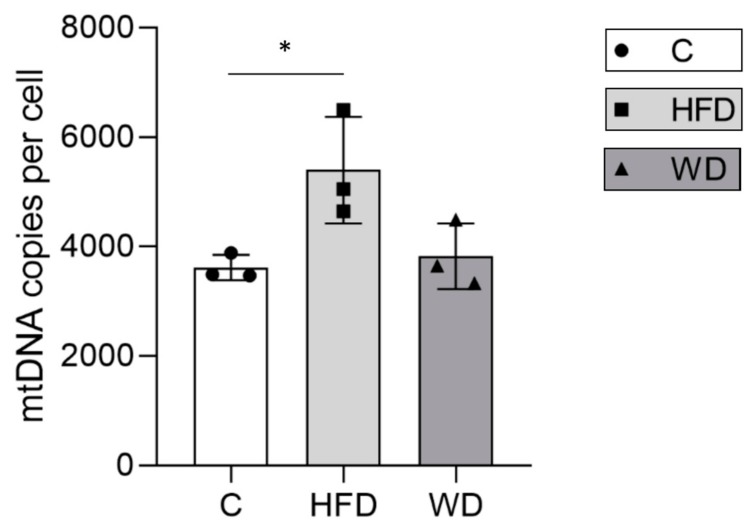
Impact of diet on mitochondrial DNA content of mouse liver. Absolute mtDNA content per cell was measured in liver of mice fed on a standard chow (C, n = 3, circle and white bar), high-fat (HFD, n = 3, square and light grey bar), and Western (WD, n = 3, triangle and dark grey bar) diets at 16 weeks. mtDNA content per cell is plotted for individual animals with error bars for mean ± SD. Significance indicated by * *p* = 0.0372 using two-tail unpaired t-test on log-transformed data.

**Figure 4 cells-08-01222-f004:**
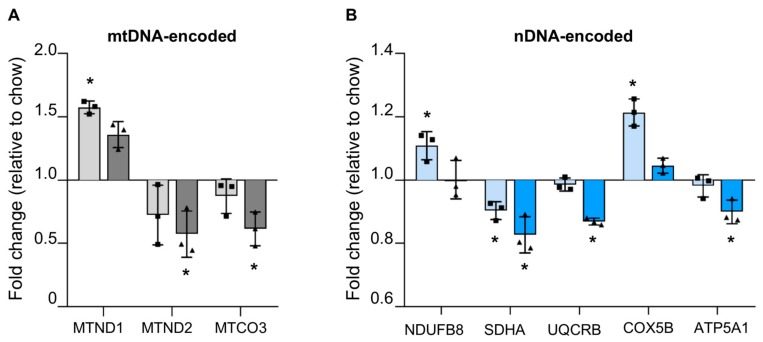
Altered levels of oxidative phosphorylation (OXPHOS) subunits in steatotic liver. The levels of several (**A**) mtDNA-encoded and (**B**) nDNA-encoded OXPHOS subunits are seen to be significantly altered in steatotic livers of mice fed with high fat (light grey/light blue) or Western (dark grey/dark blue) diets. Fold-changes are shown relative to levels in chow-fed mice. Bars indicate mean ± standard deviation for n = 3 mice. * *p* < 0.05 as determined by unpaired student t-test versus chow diet.

**Figure 5 cells-08-01222-f005:**
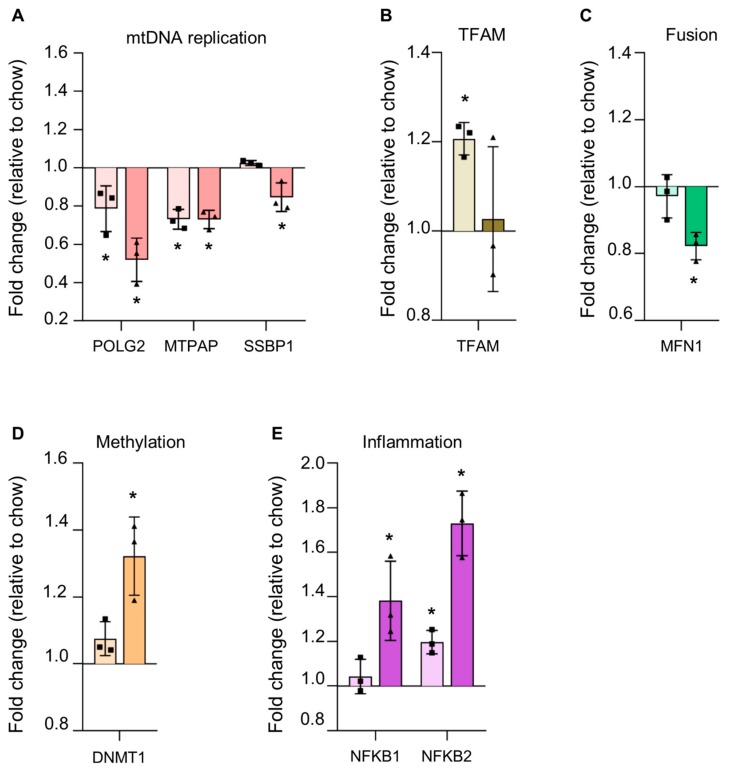
Changes to mtDNA replication, DNA methylation, and inflammation systems in steatotic liver. Altered levels of proteins involved in (**A**) mtDNA replication, (**B**) mtDNA packaging and transcription [mitochondrial transcription factor A (TFAM), (**C**) mitochondrial fusion dynamics, (**D**) nuclear genomic DNA methylation, and (**E**) inflammatory response pathways are seen in steatotic livers of mice fed with high fat (light bars) or Western (dark bars) diets. Fold change in protein level relative to chow diet fed mice is shown with bars for mean ±standard deviation. * *p* < 0.05 as determined by unpaired student t-test versus chow diet.

**Figure 6 cells-08-01222-f006:**
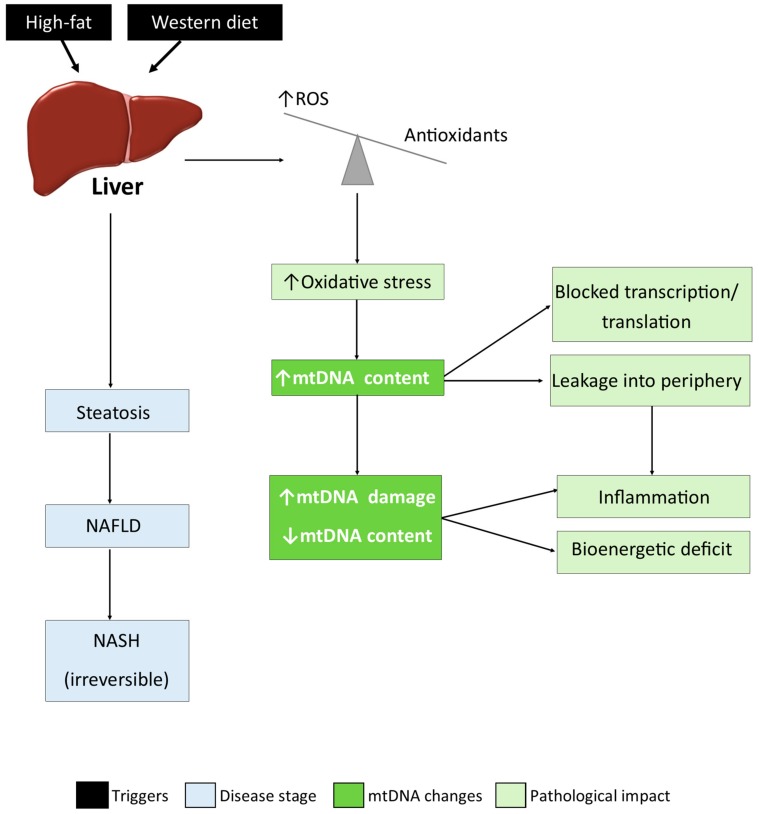
Schematic of hypothetical mechanisms for involvement of mtDNA damage in the pathogenesis and progression of non-alcoholic fatty liver disease (NAFLD).

**Table 1 cells-08-01222-t001:** Determination of absolute mitochondrial DNA (mtDNA) copy numbers per cell from mouse liver samples.

Diet Group	Animal	mtDNA Copies	*B2m* Copies	MtN	mtDNA Content Per Cell	Group mtDNA Content
Control	1	25,937,328 (1,366,548)	13,353 (979)	1942	3885	3617 (233)
	2	9,627,784 (308,494)	5512 (67)	1747	3493	
	3	23,219,751 (1,539,598)	13,377 (1,034)	1736	3472	
HFD	4	41,352,736 (2,450,499)	17,816 (704)	2321	4642	5395 (975)
	5	21,168,369 (1,697,161)	6518 (353)	3248	6496	
	6	52,091,806 (833,201)	20,641 (798)	2524	5047	
WD	7	23,516,887 (576,445)	12,888 (594)	1825	3650	3822 (600)
	8	14,047,757 (169,535)	8446 (163)	1663	3327	
	9	23,185,769 (93,493)	10,329 (461)	2245	4490	

Mice were fed one of the three diets for 16 weeks, after which total DNA was isolated and used to measure mtDNA and nuclear genome copy numbers using real time qPCR. mtDNA content per cell was calculated by dividing mitochondrial DNA copy number by nuclear DNA (*B2m*) copy number and multiplying by two to account for the diploid genome. All data are shown as mean (standard deviation).

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
