# Peer review of "A Diet Induced Maladaptive Increase in Hepatic Mitochondrial DNA Precedes OXPHOS Defects and May Contribute to Non-Alcoholic Fatty Liver Disease"

_cells, 2019, doi:10.3390/cells8101222_

Round 1

Reviewer 1 Report

The manuscript by Malik at al. focuses on the alterations in mtDNA in two models of fatty liver, one based on mice being fed a high-fat diet and the second one, closer to the so-called Western diet, when mice were fed a high-fat, high-sugar diet. The two different models are important to understand the role of dietary composition on the progression of non-alcoholic fatty liver disease to non-alcoholic steatohepatitis.

The work is very relevant as it is one of the first measurements of absolute mtDNA copy number in the liver of C57BL/6 mice under a control diet and after being fed the two diet conditions. The differences in the outcome of the two diets are actually very interesting and suggest that high-sugar may act as a second-hit stimulus which, in top of the high fat, mat accelerate of mitochondrial deterioration and inflammation in the hepatocyte. I believe this is a carefully done study, but I recommend performing some changes which will greatly improve the manuscript and increase its impact.

1.       Please change data presentation in all graphs to in Figure 3. This way, all data points are available to the reader.

2.       Description on the statistical analyses methodology used is absent from the materials and methods section. This should be added to the revised version.

3.       More details on the analysis of proteins by MS should be provided in order to allow reproductivity.

4.       Since whole tissue protein levels for Mfn1 (and not in the are presented, the authors should be careful in making assumptions about the fusion/fission balance.

5.       The authors discuss the possible role of increased DNA methylation in the WD diet for decreased gene expression in the liver but little is advanced on the potential mechanisms. Furthermore, results on other DNA methylases and demethylases is not present. Solid conclusions with only one protein analyzed are difficult to obtain. Furthermore, global DNA methylation would help strengthen the conclusions.

6.       It is not clear from the graphs whether the two diets were compared among themselves. If not, why wasn´t this done?

7.       Is there any evidence that ATP levels in control vs HF vs. HFHS groups is different?

8.       Page 10, line 316 – “since DNA in the wrong place” – this sentence should be written in a more scientific manner.

9.       Please discuss the mechanisms by which high sugar may accelerate mitochondrial deterioration in the hepatocyte, likely causing inflammation.

Reviewer 2 Report

The present manuscript deals with the interesting issue of mitochondria dysfunction in non-alcoholic fatty liver disease (NAFLD). In particular, Authors aim to evaluate the impact of diet on liver steatosis, hepatic mtDNA content and levels of key mitochondrial proteins.

To these aims C57BL/6 male mice were fed with: i) standard chow, ii) high-fat and, iii) western diet.

By comparing the absolute amount of hepatocytic mtDNA with the level of steatosis and fibrosis Authors suggested an early increase in the absolute amount of mtDNA in steatotic liver followed by reduction with the progression of liver damage.  Increase of mtDNA was associated with down regulation of nuclear and mitochondrial encoded OXPHOS proteins and proteins involved in mtDNA replication.

Authors speculate that an early maladaptive change in hepatic mtDNA content may cause subsequent bioenergetic deficit.

Finally, Authors show alterations in steatotic livers proteins (i.e. DNA (cytosine-5)-

Methyltransferase 1 nd NFKB2) suggesting altered nuclear genome methylation and increased inflammation.

Comments

Although well written, the manuscript is merely descriptive, and the reported results are not-novel.

Previous study already showed that patients with nonalcoholic steatohepatitis (NASH) featured higher mitochondrial mass, which associated with mitochondrial uncoupling, augmented hepatic oxidative stress and oxidative DNA damage paralleled by reduced anti-oxidant defense capacity and increased inflammatory response (Koliaki et al 2015). Moreover, plasma from mice and patients with NASH contains high levels of mtDNA and intact mitochondria and has the ability to activate TLR9. TLR9 pathway provides a link between the key metabolic and inflammatory phenotypes in NASH (Garcia-Martinez et al 2016).

Reviewer 3 Report

The present work proposed that changes to liver mitochondrial DNA (mtDNA) are an early event that precedes mitochondrial dysfunction and irreversible liver damage. To test this hypothesis, the authors evaluated the impact of two different diet (high fat diet and western diet, rich in fat and sugar) on liver steatosis, hepatic mtDNA content and levels of key mitochondrial proteins by quantifying absolute mtDNA copy number/cell using quantitative PCR and proteomics screen, respectively. In the western diet fed mice liver dysfunction was accelerated alongside downregulation of mitochondrial OXPHOS and replication machinery, and upregulation of mtDNA-induced inflammatory pathway. The authors stated that these results support the hypothesis that an early maladaptive increase of mtDNA may cause subsequent bioenergetic deficit and inflammation. The works address an interesting topic to clarify the mechanisms involved in NAFLD progression, however I have some major concerns about the experimental approach and the conclusion.

Major revisions:

In my opinion a main criticism for the present research work is the low number of animals for each group: n=3. For statistical analysis in research animal models, it is generally used a higher number of animals for each group (at least 6/8 animals for group), due to the high variability in the individual response to external stimuli such as diet. A higher number of animals for each group could increase the statistical significance for some protein levels that in table S1 show a no significant tendency to change. Moreover, how many animals for groups have been analysed for histological analysis? A further concern regards the lack of measurements of mitochondrial function and oxidative stress. In my opinion the conclusions of the work regarding bioenergetic deficit and ROS/antioxidant balance are not fully supported by the experimental data. In line 92-95 the authors stated that “In the current paper our aim was to determine the normal range of mtDNA copy number in mouse liver, to examine whether high fat (HFD) or Western diet (WD), … can affect this normal range during the development of  steatosis, and whether such changes correlate with changes in mitochondrial function”. No measurements of mitochondrial function and bioenergetic have been performed in the experimental work. Proteins levels measurement can be used to have information about proteins content, but it can not give a real information about protein functionality. Moreover, measurement of oxidative stress (markers of ROS production and/or antioxidant systems) should be provided to support the hypothetical mechanisms for involvement of mtDNA damage in NAFLD progression as proposed in figure 6. The authors used two different diets to address the mechanisms of NAFLD progression. I think that a time-dependent experimental model could have been a more appropriate approach to study the disease progression.

Minor revisions:

The introduction and discussion can be improved by adding further references on mitochondrial involvement in steatosis development in rodents model. Lines 97-98: the protocol number and the date of Ethical Committee approval should be added. Lines 111-112: which kind of anaesthetic was used? Lines 105-109: a table with further details on the different diets should be added. Did HFD and WD have the same composition in fatty acids in terms of saturated, monounsaturated and polyunsaturated? The different degree of liver damage may be due to a different amount of food intake. Did the authors check for food intake and body weight gain? Table S1: In the last column regarding fold change, some data for WD are lacking (Nuclear DNA-encoded OXPHOS proteins)

Round 2

Reviewer 2 Report

The revised version of this manuscript appears improved. I still feel it remains a descriptive work with interesting and nice hypothesis. Accordingly, conclusions should be tuned down.

See my comments:

I agree that providing absolute mtDNA copy numbers in normal mouse liver is undoubtedly a merit. Nonetheless, this information adds to the well-known concept that liver is a high energy requiring tissue. In fact, hepatocytes have an elevated number of mitochondria, mtDNA /nuclerDNA ratio is high (although not precise this ratio gives a good idea of the amount of mtDNA present in a certain cell), and liver is often involved in genetically determined mitochondrial disease.

Merit of this work is also the recognition that fat diet leads to mtDNA increase in the mouse liver in a short space of time. This phenomenon may be the consequence of an increase oxidative stress in the hepatocytes, as reported by Koliaki et al 2015. It is well known that increased reactive oxygen species production, in addition to damaging cells, may also signal mitochondrial biogenesis and mitochondrial DNA content. However, the link between increase oxidative stress and increase mtDNA amount is not provided in the present work. Other factors, such as mitophagy rate and mitochondrial turnover cannot be excluded and should be mentioned. Moreover, and more importantly, the link between fat diet and increase oxidative stress is not investigated. A sentence should be added highlighting these limits.

Authors also demonstrated clearly a decrease of several mitochondrial proteins, suggesting that the increase of mtDNA amount is maladaptive. However, ATP measure or biochemical analysis of respiratory chain enzymes are not provided. Moreover, mtDNA damage is supposed but not demonstrated. For example, are there multiple mtDNA deletions, as usually observed in increased oxidative stress conditions? If additional experiments cannot be provided a sentence should be added explaining the limits of the work.

Finally, I would delete the sentence “….These results support the hypothesis that an early maladaptive increase of mtDNA may cause subsequent bioenergetic deficit and inflammation via the TLR9 pathway”. In my opinion, the increase of mtDNA by itself is not the cause of the postulated bioenergetic deficit. Instead, the increased oxidative stress may trigger the activation of mitochondrial biogenesis and at the same time may damage mtDNA and mitochondrial proteins. Very interesting would be to demonstrate increased circulating broken mtDNA in patients at early stages of NAFLD, as Authors state.

Reviewer 3 Report

The authors did some work to answer to the criticisms. However, the main criticisms are still persistent and I think the manuscript is not acceptable in the present form.

Major revisions:

-The main criticism concerning the low number of animals for each group is still persistent. Moreover, the statistical analysis used in each table/graph seems to be not adequate. In the statistical analysis section, the authors stated that student t-test and one-way ANOVA were performed. However, in the figure/table captions is indicated unpaired t-test. One-way ANOVA with post-hoc test should be applied for each result. The use of one-way ANOVA analysis followed by post-hoc test could give different significance or not show significant difference between groups, in contrast with the results obtained with t-test. 

-The main discussed results  regard the different level of steatotic progression in a time dependent scale, whereas in the experimental design two different diets were used. The effect of different dietary compounds (for example fructose) should be further considered in the discussion as well as in the hypothesized mechanism and conclusion.

-The hypothesized scheme in the conclusion seems to be too speculative compared to the experimental results reported. The oxidative stress and/or anti-oxidant systems should be assessed.
